# Differentiating Induced Pluripotent Stem Cells into Natural Killer Cells for Adoptive Cell Immunotherapies—Comparative Characterization of Current Protocols

**DOI:** 10.3390/ijms26031107

**Published:** 2025-01-27

**Authors:** Tatiana Budagova, Anna Efremova, Natalia Usman, Diana Mokrousova, Dmitry Goldshtein

**Affiliations:** 1Research Centre for Medical Genetics, Moskvorechye Str. 1, Moscow 115522, Russia; workbtu@gmail.com (T.B.); diana-mok2000@yandex.ru (D.M.); dvgoldshtein@gmail.com (D.G.); 2Research Institute of Molecular and Cellular Medicine, Peoples’ Friendship University of Russia, Miklukho-Maklaya Str. 6, Moscow 117198, Russia; 3Dmitry Rogachev National Medical Research Center of Pediatric Hematology, Oncology and Immunology, Samory Mashela Str. 1, Moscow 117198, Russia; natalia.usman@yandex.ru

**Keywords:** adoptive NK cell therapy, induced pluripotent stem cells, xeno-free platforms

## Abstract

Cancers constitute a leading cause of mortality. Chimeric antigen receptor (CAR) cell therapies provide breakthrough solutions for various cancers while posing considerable risks of immunological side reactions. Of various cytotoxic lymphocyte subsets, natural killer (NK) cells are considered the least immunogenic. Obtaining viable NK cells with stable phenotypes in quantities sufficient for modification is technologically challenging. The candidate sources include primary mononuclear cell cultures and immortalized NK cell lines; alternatively, the clinical-grade NK cells can be differentiated from induced pluripotent stem cells (iPSCs) by a good manufacturing practice (GMP)-compatible xeno-free protocol. In this review, we analyze existing protocols for targeted differentiation of human iPSCs into NK cells with a focus on xeno-free requirements.

## 1. Introduction

Contemporary cancer management involves conventional and tailored approaches. Conventional surgery, chemotherapy, and radiotherapy are still the backbone of the vast majority of clinical protocols. Tailored approaches use high-tech diagnostic apparatus and manufacturing technologies to counteract specific pathogenetic mechanisms at the molecular and cellular levels; the modalities include gene therapies, targeted drug therapies, stem cell therapies, etc. [1,2,3]. Conventional approaches have multiple side effects associated with non-selective damage to cells and tissues of the body, whereas tailored approaches are substantively less toxic and, in many cases, uniquely efficacious [4,5,6]. Adoptive cell immunotherapy (ACI) is a promising tailored treatment option applicable to a broad spectrum of malignancies.

The aim of this manuscript is to review the contemporary methods of obtaining NK cells from iPSCs for adoptive immunotherapy and assess the compliance of these methods with GMP standards in terms of xeno-free requirements.

## 2. Adoptive Cell Immunotherapy

The concept of anti-cancer immunotherapy pursues the control, modification, and selective education of the patient’s immune system as a tool for specific defense against tumor growth [7]. Immunotherapies are supposed to ensure the thorough systemic clearance of tumor elements which significantly mitigates the risks of recurrence. The original adoptive cell immunotherapies require isolation of immune cells from the patient (or a donor) for cytokine stimulation and (optional) genetic reprogramming. The conditioned/engineered immune cells are subsequently transplanted to the patient [8,9]. CAR therapies use artificially designed transmembrane receptors expressed by genetically engineered lymphocytes, specifically binding tumor cell antigens to mediate lymphocyte activation [10]. The inherent versatility and tunable design of CARs ensures a high efficiency of tumor cell lysis in various models and settings.

### 2.1. CAR T-Cell Therapies

The majority of clinical trials in adoptive immunotherapy, amounting to over 500 records at ClinicalTrials.gov as of the time of writing, involve CAR T cells [11]. The U.S. Department of Health & Human Services has approved two CAR T-cell drugs for B-cell lymphoma (axicabtagene ciloleucel and tisagenlecleucel) [12]. Since 2017, the U.S. Food and Drug Administration has approved six CAR T-cell drugs (axicabtagene ciloleucel, tisagenlecleucel, brexucabtagene autoleucel, lisocabtagene maraleucel, and idecabtagene vicleucel directed against CD19-positive lymphoid tumors, and ciltacabtagene autoleucel directed against BCMA-positive multiple myeloma) [13,14].

Despite the clinical promise, CAR T-cell therapies have uncertain utility in solid cancers with cytotoxic tumor microenvironments and stromal barriers [10,12,15]. Also, CAR T lymphocytes may show reactivity to healthy tissues and cells of the host, or tumor cells may evolve alternative profiles of surface markers, which undermines the net efficacy in clinical settings. A crucial shortcoming of CAR T-cell technologies is the lack of possibility for using allogeneic T cells due to the emphatically friend-or-foe nature of T-cell immunity. The strictly autologous transplantation requirement restricts the technical opportunities for obtaining sufficient quantities of cellular product in short order [16,17]. On top of that, infusions of genetically engineered T cells may trigger severe graft-versus-host reactions entailing critical organ failure in patients with an immunocompromised status post-chemotherapy, and a neurotoxic side effect of the transfusion may cause encephalopathy [18,19]. To some extent, these drawbacks can be manipulated by using NK cells, a less immunologically conspicuous but still reliably cytotoxic lymphocyte lineage, instead of T cells.

### 2.2. CAR NK Cell Therapies

Unlike T cells, NK cells represent innate immunity, which means the lack of antigen-specific receptors similar to the T-cell receptor, and hence the possibility of allogeneic therapy. Importantly, NK cells are inherently cytotoxic toward tumor cells and capable of crossing stromal barriers, which advocates their use in solid cancers [20,21]. The natural anti-tumor cytotoxicity of NK cells can be enhanced through genetic engineering with CAR-encoding constructs [22]. Despite the promise of NK cell immunotherapy, there are relatively few corresponding clinical trials currently completed or underway [11].

The source of NK cells and methods used for their ex vivo maintenance are key parameters of NK cell-based biomedical products. The cultures can be primary, derived from mononuclear cells (MNCs) of the peripheral or umbilical cord blood. Alternatively, the cultures can be derived from humaniPSCs via controlled differentiation. The immortalized cell lines NK-92, NKG, KHYG-1, NK-YS, YT, YTS, NK3.3, NKL, IMC-1, or SNK-6 can be also considered as a substratum [23,24,25].

The listed prospective sources of NK cells for immunotherapy have specific advantages and disadvantages. Although leukapheresis-based protocols appear more straightforward than stem-cell differentiation, the MNC-derived NK cell products are heterogeneous and highly variable in terms of surface receptor signatures depending on the donor identity. In addition, the yields of NK cells by leukapheresis are insufficient for a standardized biomedical cell product, which further limits the clinical value of the approach. A standard single donation provides 350–500 mL of peripheral blood. About 10^5^ NK cells per mL of donated blood can be harvested, about 5% of which will express the CD56^bright^ phenotype preferable for anti-cancer therapy [26]. The bulk CD56^dim^ population, although highly cytotoxic, is prone to quickly losing this capacity ex vivo, as the densities of key activating surface receptors, notably CD16, tend to decline [24,27]. In current trials, the patients are administered varying doses of NK cells ranging from millions to billions [11]. In one study, CD56+CD122+ rates constituted 8% (median) of 24.3 × 10^9^ (median) MNCs obtained by leukapheresis. After the depletion of CD3+ cells, MNC counts decreased to 15.4 × 10^9^ (median) and CD56+CD122+ rates reached 29% (median). Even so, this occurred with a maximum 3–7-fold expansion of the NK cell isolate [28].

Immortalized NK cell lines are stable continuous cultures derived from NK/T-cell lymphomas. Despite the fact that such lines rapidly proliferate in culture and can be conveniently manipulated, the majority exhibit low cytotoxicity levels. In addition, the cells must be irradiated before administration to a patient in order to suppress division, which negatively affects their functional capacities and confines the lifespan of the transplant to 48 h. To give an example, for the NK-92 cell line, the minimum radiation dose suppressing division while sparing functionality is 1000 cGy [25,29].

Human iPSCs are considered a promising source of NK cells for immunotherapy. Directed differentiation of iPSCs into NK cell lines provides homogeneous stable cultures that are easy to handle and scale up for clinical use; importantly, the cells can be genetically engineered at early stages of differentiation [30,31]. The yield of the final cell product is unlimited and the problem of donor selection is eliminated by the autologous origin of iPSCs. Thus, the iPSC-derived NK cells make a strong candidate substratum for immunotherapy [29]. Registered clinical trials of iPSC-derived CAR NK cell products are listed in Table 1.

Despite the very encouraging experimental data, obtaining a biomedical NK cell product from iPSCs is complicated by the lack of unified protocols compatible with current GMP in terms of xeno-free requirements, i.e., conditions for culturing human cells without the use of animal products. The protocols for directed differentiation of iPSCs into NK cells remain extremely diverse and focused on narrow goals set by individual research teams. Consequently, the immediate prospects of using iPSCs as a source of clinical-grade NK cells are still narrow.

## 3. Developmental Perspective

Despite the diversity, protocols for obtaining NK cells from iPSCs employ the authentic principles of NK cell differentiation in the body. The major site of NK cell development is the bone marrow, although some NK cells develop in the tonsils, thymus, liver, spleen, and lymph nodes [32,33].

NK cell lineages constitute a distinct hemopoietic progeny central to the subject of this review. The ultimate progenitors termed hemopoietic stem cells are converted consecutively into common lymphoid progenitors (LPs), early and late NK cell precursors, immature NK cells, and finally, mature NK cells. The whole sequence of the transitions is controlled by a spatiotemporal network of transcription factors, second messengers, phosphorylation cascades, cytokines, and chemokines; the division into stages reflects specific signatures of cell surface markers (Figure 1) [33,34]. Understanding the dynamics of surface markers and their interactions with various ligands allows us to tune the protocols toward higher yields and stronger functionalities of the iPSC-derived NK cells. The immature NK cells of CD56^bright^ phenotype are a particularly desirable substratum for biomedical cell products due to the incremental expression of multiple activating receptor types at this stage. Once settled, cells of a particular phenotype can be expanded exponentially, especially under stimulation with mitogenic ligands. At the stage of mature NK cells, the product reaches maximum cytotoxicity, which is a transient state, as the expression of activating receptors reaches a plateau and then declines.

Expression of a particular surface receptor determines interactions with corresponding molecular elements of the cellular microenvironment that direct and regulate further development, most notably interleukin (IL) 2, IL7, IL15, IL21, SCF, and Flt3-L. Of these, IL7 determines the conversion of LPs to early NK precursors, marked by the onset of surface expression of its receptor CD127. IL7 also promotes survival of the “elite” immature NK cells with the CD56^bright^ phenotype through increased expression of anti-apoptotic protein BCL-2, albeit without a boost in cytotoxicity and/or switch to the CD56^dim^ phenotype. Of the other factors, IL2 and IL15 are key to the transition from early to late precursors, which is marked by the expression of CD122 acting as a subunit in the receptors of both cytokines. The ability of NK cells to interact with IL2 and IL15 is directly related to Flt3L signaling, which controls the expression of β-chain in CD122. Combined with SCF, Flt3L also facilitates the conversion of hemopoietic stem cells into LPs. The role of IL15 is perhaps the most important, as it orchestrates the development from early precursors to fully mature NK cells. In combination with other interleukins, IL15 can afford an ultra-pure population of NK cells from iPSCs; the technicalities are discussed in the next section. Experimental knockdown of IL15 receptor subunits α/β and IL15 eliminates the process of NK cell development, whereas similar inactivation of IL2/4/7 signaling does not. At the same time, IL2 and IL21 support the proliferation of activated NK cells and stimulate the formation of perforin- and granzyme-containing exosomes; the net effect significantly enhances NK cell cytotoxicity. The surface expression of CD117 by NK precursors mediates the interaction with SCF, which stimulates proliferation via a MAPK cascade. In mature NK cells, IL2 and IL15 can promote the transition of CD56^bright^ to CD56^dim^ by yet unknown molecular mechanisms, whereas IL12 and IL18 enhance NK cell cytotoxicity by inducing IFN-γ production (Figure 1) [33,34,35,36,37,38]. Lymphotoxins LTα and LTβ form LTα1β2 heterotrimers anchored to the surface of activated lymphocytes and binding LTβ receptors at the stromal cell surface. The complexes have been shown to facilitate the early stages of NK development, which explains the utility of mouse stromal cell lines, notably OP9, in NK cell differentiation protocols.

IL17 plays a dual role as immunosuppressive yet pro-inflammatory cytokine, which directly inhibits the development and functional activity of NK cells through elevated expression of SOCS3, a potent suppressor of cytokine signaling. SOCS3 inhibits IL15-induced STAT5 phosphorylation. At the same time, IL17 can indirectly stimulate the lipopolysaccharide-primed NK cells to enhance their cytolytic activity. Another immunosuppressive factor, TGF-β, inhibits NK cell development by blocking the IL15-induced mTOR signaling and activating the Smad3 cascade [38,39].

Transcription factors involved in NK cell development can be divided into factors that participate in differentiation from LPs and factors that participate in the CD56bright-to-dim phenotype transition (maturation; Table 2). Transcription factors crucial for NK cell development include E4BP4, TOX, ID2, T-bet, EOMES, and ZEB2 [40].

E4BP4 marks the differentiation from LPs to immature NK cells; noteworthily, E4BP4 is not expressed by T- or B-cell lineages. E4BP4 is considered a master regulator, as its deficiency cannot be compensated for by EOMES, ID2, or T-bet, despite influencing their expression. A major cellular target of E4BP4 is NOTCH1. The inactivation of NOTCH1 inhibits differentiation of NK cells from LPs, but does not interfere with subsequent maturation (i.e., transition from CD56^bright^ to CD56^dim^ phenotype). Mechanistically, NOTCH proteins form complexes with Jagged and Delta to act as transcription activators. NOTCH proteins are crucial for T-cell differentiation in the thymus. The Rbp–JK interaction of the NOTCH signaling cascade can stimulate directed differentiation of NK cells from precursors and boost mature NK cell cytotoxicity [35,36,37,39,40,41].

The transcription factor TCF1, originally implicated in T-cell development, is expressed by NK cell lineages to orchestrate the transition from early to late precursors, but not their subsequent maturation. Similarly, the transcription factor ETS1 is expressed until the late precursor stage; its targets include the transcription factors T-bet, ID2, GATA3, and BLIMP1. ETS1 expression, stimulated by IL2 and IL15, has been shown to regulate the transition from late precursors to immature NK cells [35,36,37,39,41].

The JAK/STAT molecular cascade is essential for NK cell maturation. The kinases JAK1,3 are activated upon binding of IL15 to its receptor. The activated JAK1 and JAK3 bind, respectively, IL-2Rβ and IL-2Rγc, which leads to STAT3,5 phosphorylation.

The conversion of LPs to NK cells also involves the transcription factors ID, ID2, and ID3. The binding efficiency of IL15 to its receptor is ID2-dependent. The experimental inhibition of ID2 does not affect the number of LPs, but negatively affects their conversion to NK cells at all stages [37,39,41].

NK cell maturation, defined as the transition from the CD56^bright^ to the CD56^dim^ phenotype, requires the transcription factors T-bet and EOMES. T-bet directly regulates S1pr5, and its deficiency affects the migration of NK cells from the bone marrow. EOMES is also necessary for the maintenance of mature status by NK cells. However, a deficiency of T-bet or EOMES does not entail a reduction in the functional capacities of NK cells at the body level.

Overall, the CD56bright-to-dim transition (maturation) is a complex process orchestrated by multiple transcription factors, as listed in Table 2. The inhibition of TOX and TOX2 affects not only the counts of mature NK cells, but also the expression of several other transcription factors (T-bet, ID2) involved in maturation [37,39,41]. The cytokine IL15, the master inducer of NK cell development, is produced by dendritic cells, monocytes, and macrophages. A major portion of IL15 is presented to NK cells and their precursors in an anchored state, cis-bound to the IL15 receptor α subunit (CD122). The complexes bind β and γ subunits at the NK cell surface, encoded by the *IL2RB* gene expressed under control of RUNX3 and EOMES in immature and mature NK cells, respectively. The interaction triggers several signaling cascades: firstly, the JAK-STAT5 pathway. The IL15-induced phosphorylation of STAT5a/b by JAK1/3 leads to di- or tetramerization of the STAT proteins. The complexes are translocated to the nucleus to regulate the activity of multiple transcription units, notably MCL1, BCL2, VEGFA, MYC, NFKB1, IFNG, and TNF, thereby supporting cell proliferation and survival while suppressing apoptosis [42,43]. Secondly, IL15 activates the PI3K-Akt cascade. PI3K promotes conformational changes in Akt leading to Akt phosphorylation by PDK1 and ultimately mTORC1 and mTORC2 activation. Of these, mTORC1 via E4BP4 via EOMES activates the *IL2RB* gene encoding the β and γ subunits of CD122, thereby supporting the sensitivity of NK cells to IL15 and IL2; mTORC2 via Akt inhibits FOXO1 functionality and via GATA3, TOX2, and ETS1 proteins activates the expression of T-bet [42]. Thirdly, the Sch-Grb2 pathway is activated, which is upstream of the Ras-Raf-MAPK and PI3K-Akt cascades, and accordingly further entrenches the effect of higher survival and lower apoptosis rates [43,44].

Each stage of differentiation/maturation can be identified by specific expression profiles of surface protein molecules. Key surface markers for each stage are presented in Table 3 and Figure 1 and Figure 2. In the case of CD117-negative LPs, it is necessary to take into account that the late precursor stage is also CD117-negative. CD34 and CD133 provide universal positive markers throughout the differentiation. The CD244 marker starts to appear at the stage of LPs and persists until the stage of mature NK cells. The CD7 marker appears at the stage of early NK cell precursors and persists into the latest stage of NK cell differentiation. CD122 has a similar temporal pattern. The CD127 marker can be used to identify NK cell precursors. Immature NK cells start expressing NKP46, NKP30, and CD161 receptors which persist into the stage of mature NK cells. The activating NKG2D receptors start to appear at stage 4a of immature NK cells, and so do the inhibitory NKG2A receptors. Importantly, the majority of activating receptors are expressed starting from late stages of differentiation. CD57 becomes a positive marker as late as during the full maturation stage [32,33,34,45,46]. The use of CD3 and CD19 markers of T and B cells, respectively, should be mandatory in order to exclude the differentiation of iPSCs into alternative (non-NK) lymphocyte lineages.

## 4. Comparative Analysis Protocols

Before considering the various approaches to the directed differentiation of human iPSCs into NK cells, it should be emphasized that xeno-free requirements for cell isolation, maintenance, engineering, and expansion are essential in developing a cell product for solid tumor therapy. The term “xeno-free” means that all components must be either of inorganic nature including those synthesized chemically, or isolated from cells/tissues of the same species (human in the case of biomedical products). The use of biological material from other species, particularly animal, at any stage of the production of a drug (cell product) for internal administration creates the risks of contamination with foreign antigens and pathogens causing extra immunological/infectious burden and toxicity reactions [47,48].

All current protocols for NK cell differentiation from iPSCs mimic the stages of NK cell development in the body and mostly involve two principal steps (Figure 2):Differentiation of stem cells into LPs;Directed differentiation of LPs into NK cells.

Several current protocols for iPSC differentiation into NK cells use feeder and stromal cells to obtain first LPs and then the final product. Specifically, most protocols published roughly before 2015 used feeder and stromal cells, as well as xenobiological supplements, since the main goal was to develop a methodology for obtaining phenotypically pure NK cells in high yields. Later on, the developments became focused on fulfilling the xeno-free standard. To illustrate the methodological progress in obtaining NK cells from iPSCs, several protocols will be considered as examples below, starting with a feeder-based protocol. The methods are summarized in Table 4.

NK cell lineages constitute a distinct hemopoietic progeny central to the subject of this review.

### 4.1. Method #1

A seminal chapter entitled “Human pluripotent stem cells as a renewable source of natural killer cells” by Hermanson, Ni, and Kaufman (2015) describes a two-step approach starting with the formation of embryoid bodies (EBs) in order to boost the yield of phenotypically stable LPs [49,50,51].

Trypsin adaptation is used as a pre-conditioning step to support the maintenance of iPSCs in single-cell suspensions prior to EBs formation. The step involves mouse embryonic fibroblast (MEF) lines as a feeder; the subsequent EB formation proceeds on an MEF feeder in BPEL culture medium (Iscove’s modified Dulbecco’s medium (IMDM) supplemented with bovine serum albumin (BSA), recombinant human (rh) SCF, rhBMP4, and rhVEGF; Table 4). The formation of EBs comprising 20–40% LPs (CD34+CD45+) takes 11 days. The second step, differentiating the progenitors into NK cells, involves the irradiated stromal cell line EL08-1D2 grown on gelatin and in high-glucose Dulbecco’s modified Eagle medium (DMEM) supplemented with heat-inactivated human AB serum, rhIL3, rhSCF, rhIL7, rhIL15, and rhFLT3L. The step proceeds for 4–5 weeks with the medium replaced every 5–7 days. Thus, the protocol affords NK cells from iPSCs in about 7–8 weeks (without trypsin adaptation—5–6 weeks). Originally, 3 × 10^6^ to 10^7^ iPSC-derived NK cells were harvested from two 24-well plates (~10^2^ cm^2^ total surface area).

Subsequent expansion and activation are carried out in RPMI-1640-based medium supplemented with fetal bovine serum (FBS) and recombinant IL2. A modification of the K562 chronic myeloid leukemia-derived cell line [51], irradiated, is used as a feeder in a decreasing proportion to the cells: 2:1 (K562-mbIL-21 feeder to NK, respectively) on week 1, and 1:1 later on. The expansion proceeds for over 60 days; the phenotypes are analyzed by flow cytometry and the scaling procedure affords an over 500-fold expansion to a final yield of 10^9^ cells in the original setting.

Despite the reproducibility, cost-efficiency, and high yields of the pure NK cell product, and the use of “clean” recombinant human factors, the method better suits research purposes than clinical applications due to the use of animal-derived supplements, notably the feeders (MEF, EL08-1D2) and BSA in BPEL culture medium. The MEF cell line is most commonly used for culturing undifferentiated human iPSCs to achieve stable growth. To fulfil the xeno-free requirements, the MEF cell line should be substituted with human analogues, e.g., human embryonic fibroblasts or urogenital tract epithelial cells. Alternatively, undifferentiated iPSCs can be cultured without the use of a feeder layer, as in some other protocols [52,53], though this will not eliminate the problem altogether, as the EL08-1D2 stromal cell line subsequently used for obtaining NK cells from LPs has been derived from the liver of transgenic mice. Despite the proven advantages of this cell line as a stromal support for NK cell lineages in vitro [54], considering the xeno-free requirements, it should be abandoned and possibly substituted with human splenic fibroblasts as a replacement, or conditions can be selected that do not require the use of stromal cell lines, for example, culturing in Glycostem^®^ medium with heparin which stabilizes the cytokine additives [54,55,56]. BSA, although considered non-toxic, low-antigenic, and able to enter the product only in trace amounts, can nevertheless induce a specific immune response in human recipients [57]. As a substitute, human serum albumin should be used to supplement the culture medium; a commercial solution is APEL™ [57]. Furthermore, the medium used to expand the iPSC-derived NK cells contains FBS, contrary to the xeno-free requirements.

Experiments to assess the survival of iPSC-derived NK cells under in vivo conditions have not been conducted.

### 4.2. Method #2

In 2019, Zhu and Kaufman published “An Improved Method to Produce Clinical-Scale Natural Killer Cells from Human Pluripotent Stem Cells”, describing the production of NK cells from iPSCs without using feeder cell lines [58]. Similarly with the previous version, the protocol consists of two steps: (1) obtaining embryoid bodies (EBs) enriched with LPs (CD34+CD45+) and (2) directed differentiation of LPs into NK cells (CD45+CD56+). Human iPSCs are adapted to feeder-free conditions over 1–2 passages in mTeSR1 medium, using Matrigel^®^ coating to enhance the adherence of cells to the surface of the culture plate. Importantly, for iPSCs already maintained under feeder-free conditions, the adaptation should be skipped. EBs formation was carried out in APEL™2 medium supplemented with human serum albumin, recombinant human SCF, BMP4, and VEFG, and a Rho-associated coiled-coil containing protein kinase (ROCK) inhibitor Y-27632 (Table 4). The incubation proceeds for 6 days (cf. 11 days in the 2015 protocol [49]) to yield EBs highly enriched with CD34+ LPs (>50% in the original setting). Further differentiation of LPs into NK cells proceeded in APEL medium supplemented with heat-inactivated human AB serum and recombinant IL3, SCF, IL7, IL15, and FLT3L. The culture plates were coated with pig skin-derived gelatin; the medium was replaced every 5–7 days. The differentiation is complete in 3–4 weeks, which is slightly faster than in the original protocol of 2015. The expansion of iPSC-derived NK cells is carried out using the irradiated modified cell line K562-mbIl-21 and RPMI-1640 medium supplemented with recombinant IL2. The phenotypes are analyzed by flow cytometry. The original setting yielded 2 × 10^6^ to 2 × 10^7^ NK cells from a single 6-well plate (60 cm^2^ total surface) prior to expansion, and after scaling the number reached 10^9^ cells, meaning an over 500-fold expansion.

The main advantage, the use of feeder-free systems throughout all steps, is complemented by the high efficiency of the approach and the pure yields of the cell product. Still, no straightforward translation to clinical manufacturing would be possible, as the use of Matrigel^®^ and gelatin (porcine) coatings contradicts the xeno-free requirements. Matrigel^®^, a solubilized basement membrane matrix rich in laminin-111, collagen, and growth factors, secreted by the Engelbreth–Holm–Swarm mouse sarcoma cell line, has been developed to improve cell adherence to the culture plate surface [59,60,61]. Substitutes for Matrigel^®^ can be provided by protein/polysaccharide coatings either derived from human biological material or synthetic. Natural coatings similar in properties to Matrigel^®^ include vitronectin, alginate, chitosan, dextran, cellulose, hyaluronic acid, agarose, pectin, gelatin, fibrin, collagen, poly-L-lysine, and heparin hydrogels [62]. As in the 2015 protocol, FBS is used as a supplement during the expansion step, which also contradicts the xeno-free requirements.

Experiments examining the survival of iPSC-derived NK cells in vivo have not been conducted.

### 4.3. Method #3

“Recapitulative haematopoietic development of human pluripotent stem cells in the absence of exogenous haematopoietic cytokines” by Philonenko et al. (2021) describes NK cell production from iPSCs performed with very restricted cytokine supplementation [63]. The results indicate that, with minimal amounts of exogenously added agents, endogenous stimuli in the culture can suffice to drive the transition from iPSCs to LPs and further to NK cells. It should be noted that the study was not specifically focused on NK cell lineages, but rather on a range of conditions for differentiating iPSCs into LPs with diverse potentialities, while the full differentiation into NK progeny constituted a small segment of the study.

The data emphasize the necessity of culturing iPSCs for 2–3 passages in mTeSR™1 on Matrigel^®^, apparently as a way of adaptation to feeder-free conditions. The formation of EBs from iPSCs is carried out in mTeSR1™1 supplemented with BMP4 and VEGF (recombinant human) and thiazovivin, a low-molecule ROCK inhibitor similar to Y-27632 [64]; the step is complete within 48 h and there is no provision for media replacement. In the original setting, the harvested total yields at this step were >60% CD34-positive. The next step, obtaining NK cells from CD34+ progenitors, uses α-MEM medium supplemented with the recombinant human proteins SCF, IL7, FLT3L, and β-mercaptoethanol, and the OP9-DL4 cell line as a stromal support (Table 4). The culture plate is coated with collagen IV from rat tissues. Collagen, a principal component of the extracellular matrix, can accommodate the cells in culture with a net positive effect on cell adherence, survival, and growth [64,65].

The OP9-DL4 stromal cell line is derived from mouse bone marrow; the cells express characteristic surface signatures that mimic thymic microenvironments, e.g., the NOTCH ligand Delta-like 4, thereby inducing iPSCs to differentiate into lymphoid progeny [66]. The incubation proceeds for 3–5 weeks with both the medium and the stromal support replaced every 7 days (Table 4). The phenotypes are analyzed by flow cytometry; originally, the authors also applied transcriptomic analysis of the progeny at consecutive stages of differentiation. The protocol affords cell products with over 40% NK cells (CD16+CD56+) [63].

The key distinction of Method #3 is the minimization of biologically active additives in culture media during the entire procedure. First, differentiation into LPs is carried out without adding SCF to the culture medium, and the medium supplemented with BMP4 and VEGF is eventually replaced with the medium containing VEGF as a single added biological. The next stage, differentiation per se, proceeds without added IL3 and/or IL15 in the culture medium, though uses the OP9-DL4 stromal support [63]. “Minimal” supplementation with SCF, FLT3L, and IL7 is a prerequisite for the differentiation of LPs into NK cells in vitro, even in cocultures with OP9-DL4. Alternatively, without stromal support, IL3 and IL15 should be added to the medium on top of the “minimal” formula, which is also preferable in terms of differentiation specificity: the CD56+ NK cell product will be >90% pure, whereas with the stromal support the product will contain a high proportion of T cells [26]. Thus, one important drawback of this protocol for obtaining a “pure” population of cytotoxic NK cells is the use of the mouse stromal cell line OP9-DL4, which contradicts the xeno-free requirements, and so does the use of Matrigel^®^ coating and FBS in differentiation medium. To achieve GMP compatibility, the murine stromal elements used as a source of active molecules should be substituted with a (more potent) combination of IL3 and IL15, probably boosted by exogenously administered lymphotoxins LTα and LTβ.

Experiments examining the survival of iPSC-derived NK cells in vivo have not been conducted.

### 4.4. Method #4

“Differentiation of natural killer cells from induced pluripotent stem cells under defined, serum-, and feeder-free conditions” by Lupo et al. (2021) describes a similar approach to NK cell production from iPSCs with the involvement of EBs to enhance the intermediate yields of LPs [67]. Originally, human iPSCs were cultured in mTeSR1 on Matrigel^®^ for 4–5 days as a preconditioning step. EB formation was carried out in APEL™2 supplemented with SCF, BMP4, VEGF, and a ROCK inhibitor (Table 4). The 11-day culture (the medium being replaced at 3-day intervals) afforded >50% relative yields of CD34+ progenitors in EBs. Subsequent differentiation of LPs into NK cells used APEL™2 supplemented with recombinant human SCF, IL7, IL15, FLT3L, and IL3. The culture proceeded for 4 weeks; the percentage yield of CD56+ NK cells was >30%. The original setting afforded ≥3 × 10^5^ per 10^2^ EBs per single 96-well plate. The product was subsequently scaled up in a CTS™ OpTmizer™ supplemented with human AB serum and recombinant human IL15, IL2, and IL21. The expansion and activation step proceeded for 3–4 weeks; the cells increased in number 4–5-fold; in the original setting, the expansion yielded >10^9^ viable NK cells from a single 24-well plate. The phenotypes were analyzed by flow cytometry.

The advantages of this protocol include feeder-free conditions at both differentiation steps (iPSCs to LPs and LPs to NK cells) and the use of biological additives of non-animal origin (recombinant human proteins) in the composition of the culture media. The only compromising issue in terms of xeno-free requirements is the use of Matrigel coating. The rates and yields of NK cell production from iPSCs by Method #4 are comparable with those in the other protocols considered in this section.

Experiments to assess the survival of iPSC-derived NK cells under in vivo conditions have not been conducted.

**Table 4 ijms-26-01107-t004:** Comparative analysis of protocols for obtaining clinical-grade NK cells from human iPSCs.

Ref. #	Step	Length	Culture Medium	Feeder/Stromal Support	Xeno-Free Requirements	Comments
Hermanson, D. L.; Ni, Z.; Kaufman, D. S., 2015 [49]	Trypsin adaptation	12–15 passages	Not specified.	** MEF **	Non-compliant	Trypsin adaptation is used as a pre-conditioning step to support the maintenance of iPSCs in single-cell suspensions; EB formation facilitates the conversion into CD34+CD45+ LP phenotypes with high efficiency; The use of xenogeneic feeder/stromal lines to differentiate LPs into NK cells provides higher yields, but contradicts the GMP guidelines.
	Differentiation of iPSCs into LPs via EB formation	8–11 days	BPEL: IMDM 43%, F-12 43%, **BSA** 0.25%, Polyvinyl alcohol 0.25%, 0.1 µg/mL Linoleic acid, Synthechol 1×, α-Monothioglycerol 450 µM, Protein-free hybridoma medium II 5%, 50 µg/mL Ascorbic acid 2-phosphate, ITS 1×, SCF 40 ng/mL, BMP4 20 ng/mL, VEGF 20 ng/mL.	** MEF **		
	Differentiation of LPs into NK cells	4–5 weeks, medium replaced every 5–7 days	Differentiation medium: 56.6% DMEM high glucose, 28.3% HAMS/F12, human AB serum heat-inactivated 15%, β-Mercaptoethanol 1 uM, sodium selenite 5 ng/mL, Ethanolamine 50 µM, Ascorbic acid 20 mg/L, IL-3 5 ng/mL, SCF 20 ng/mL, IL7 20 ng/mL, IL15 10 ng/mL, FLT3L 10 ng/mL.	** EL08-1D2 (or AFT024) **		
	NK cell expansion (scaling up)	Up to 60 days, medium replaced every 3–4 days	Expansion medium: RPMI-1640, **FBS** 10%, IL2 50 U/mL.	mbIL-21 K562		
Zhu, H.; Kaufman, D. S., 2019 [58]	Adaptation of human iPSCs to feeder-free conditions	3 passages	mTeSR™1 with **Matrigel^®^** coating.	Not used	Non-compliant	The step of adaptation to feeder-free conditions is optional for cells already grown under feeder-free conditions.
	Differentiation of iPSCs into LPs via EB formation	6 days	Differentiation medium: STEMdiff™ APEL™2, SCF 40 ng/mL, BMP4 20 ng/mL, VEGF 20 ng/mL, ROCK inhibitor Y-27632 10 µM.	Not used		
	Differentiation of LPs into NK cells	3–4 weeks, medium replaced every 5–7 days	Differentiation medium: DMEM + GlutaMAX™-I 56.6%, F12+GlutaMAX™-I 28.3%, human AB serum heat-inactivated 15%, β-Mercaptoethanol 1 μM, Sodium selenite 5 ng/mL, Ethanolamine 50 μM, Ascorbic acid 20 mg/L, IL3 5 ng/mL, SCF 20 ng/mL, IL7 20 ng/mL, IL15 10 ng/mL, FLT3L 10 ng/mL.	Not used		
	NK cell expansion (scaling up)	Over 90 days, medium replaced every 3–4 days	Expansion medium: RPMI-1640, **FBS** 10%, IL2 50 U/mL.	mbIL-21 K562		
Zeng, J.; Tang, S. Y.; Toh, L. L.; Wang, S., 2017 [68]	Differentiation of human iPSCs into LPs	12 days, medium replaced every 4 days	α-MEM, **FBS** 20%.	** OP9 **	Non-compliant	The protocol omits EB formation; Differentiating LPs to NK cells involves several rounds of re-plating the coculture, which distinguishes the protocol among the rest.
	Differentiation of LPs into NK cells	28 days, trypsinization and fresh medium every 7 days	α-MEM, **FBS** 20%, SCF 10 ng/mL, FLT3L 5 ng/mL, IL7 5 ng/mL, IL15 10 ng/mL.	** OP9-DLL1 **		
Philonenko, E. S.; Tan, Y.; Wang, C.; Zhang, B.; Shah, Z.; Zhang, J.; Ullah, H.; Kiselev, S. L.; Lagarkova, M. A.; Li, D.; Dai, Y.; Samokhvalov, I. M., 2021 [63]	Culture of human iPSCs	2–3 passages	mTeSR™1 with **Matrigel^®^** coating.	Not used	Non-compliant	Differentiation into LPs is induced by added VEGF plus endogenous stimuli; The data underscore the high efficiency of conversion to LPs for human iPSCs.
	Production of EB and differentiation into LPs	48 h	mTeSR™1, BMP4 2–4 ng/mL, VEGF 50 ng/mL, Thiazovivin 10 μM.	Not used		
	Differentiation of LPs into NK cells	5 weeks, medium replaced every 7 days	OP9 differentiation medium: α-MEM, **FBS** 20%, β-mercaptoethanol 50 µM, SCF 5 ng/mL, IL7 5 ng/mL, FLT3L 5 ng/mL.	** OP9-DL4 **		
Lupo, K. B.; Moon, J.-I., 2021 [68]	Culture of human iPSCs	4–5 days	mTeSR™1 with **Matrigel^®^** coating.	Not used	Non-compliant	
	Production of EB and differentiation into LPs	11 days, medium replaced every 3 days	Differentiation medium: STEMdiff APEL™2, SCF 40 ng/mL, BMP4 20 ng/mL, VEGF 20 ng/mL, ROCK inhibitor Y-27632 10 µM.	Not used		
	Differentiation of LPs into NK cells	4 weeks	Differentiation medium: STEMdiff APEL™2, SCF 20 ng/mL, IL7 20 ng/mL, IL15 10 ng/mL, FLT3L 10 ng/mL, IL3 5 ng/mL.	Not used		
	NK cell expansion (scaling up)	Up to 3–4 weeks	Expansion medium: CTS OpTmizer™ human AB serum 5%, IL15 10 ng/mL, IL2 500 IU/mL, IL21 25 ng/mL	Not used		
Goldenson, B. H.; Zhu, H.; Wang, Y. M.; Heragu, N.; Bernareggi, D.; Ruiz-Cisneros, A.; Bahena, A.; Ask, E. H.; Hoel, H. J.; Malmberg, K.-J.; Kaufman, D. S., 2020 [69]	Production of EB and differentiation into LPs	8 days	Differentiation medium: APEL™2, SCF 40 ng/mL, VEGF 20 ng/mL, BMP4 20 ng/mL.	Not used	Compliant	The protocol uses trypsin-adapted human iPSCs.
	Differentiation of LPs into NK cells	28–32 days	Differentiation medium: IL3 5 ng/mL, IL15 10 ng/mL, IL7 20 ng/mL, SCF 20 ng/mL, FLT3L 10 ng/mL.	Not used		
	NK cell expansion (scaling up)	-	-	K562-IL21-41BBL		
Lupo, K. B.; Yao, X.; Borde, S.; Wang, J.; Torregrosa-Allen, S.; Elzey, B. D.; Utturkar, S.; Lanman, N. A.; McIntosh, M.; Matosevic, S., 2024 [70]	Culture of human iPSCs	4–5 days	mTeSR™1 with **Matrigel^®^** coating.	Not used	Non-compliant	
	Production of EB and differentiation into LPs	11 days, medium replaced every 3 days	Differentiation medium: APEL™2, SCF 40 ng/mL, VEGF 20 ng/mL, BMP4 20 ng/mL, ROCK inhibitor Y-27632 10 µM.	Not used		
	Differentiation of LPs into NK cells	4 weeks, fresh medium twice a week	Differentiation medium: APEL™2, IL7 20 ng/mL, IL15 10 ng/mL, SCF 20 ng/mL, FLT3L 10 ng/mL, IL3 5 ng/mL.	Not used		
	NK cell expansion (scaling up)	2–3 weeks	Expansion medium: CTS NK Xpander™, human AB serum 5%, IL2 500 IU/mL.	Not specified		

NK, natural killer; iPSCs, induced pluripotent stem cells; EB, embryoid bodies; FBS, fetal bovine serum; (rh)SCF, (recombinant human) stem cell factor; (rh)IL#, (recombinant human) interleukin; (h)VEGF, (human) vascular endothelial growth factor; (h)BMP4, human bone morphogenetic protein 4; α-MEM, minimal essential medium Eagle—alpha modification; MEF, mouse embryonic fibroblast culture (immortalized); BPEL, APEL™ medium supplemented with bovine serum albumin instead of recombinant human albumin; IMDM, Iscove’s modified Dulbecco’s medium; (rh)FLT3L, (recombinant human) Fms-like tyrosine kinase 3 ligand; ITS, Insulin–Transferrin–Selenium supplement; DMEM, Dulbecco’s modified Eagle medium; ROCK, Rho-associated coiled-coil containing protein kinase; LPs, lymphoid progenitors. Components that violate the xeno-free terms are marked in bold and underlined.

Despite the fact that the protocols described in this work do not provide results on the survival of iPSC-derived NK cells in vivo, it is known that iNK cells retain their viability upon grafting to laboratory animals and exhibit high cytotoxicity against tumor cells. This has been demonstrated in both research papers and the clinical trials conducted to date: NCT03841110, NCT04630769, NCT05182073, NCT06245018, and NCT06702098.

Additional examples may be found in the following papers:-In 2013, D.S. Kaufman developed the protocol for obtaining mature NK cells from iPSCs, performing a thorough analysis of transplanting obtained NK cells to immunodeficient mice and visualizing their survival with K562 tumor cells in his paper, “Development, expansion, and in vivo monitoring of human NK cells from human embryonic stem cells (hESCs) and induced pluripotent stem cells (iPSCs)” [71].-In the 2020 study “Pluripotent stem cell-derived NK cells with high-affinity non-cleavable CD16a mediate improved antitumor activity”, the D.S. Kaufman team generated genetically modified iPSCs with for their further differentiation into NK cells with high CD16 expression. The resulting hnCD16-expressing NK cells were injected to immunodeficient mice and their survival and cytotoxicity against Raji multiple myeloma cells was assessed. The authors showed that iNK cells not only maintained their viability in vivo, but also effectively eliminated the tumor cells in mice when they were co-administered with CD20 antibodies [72].-In 2020, the D.S. Kaufman team in their study “Metabolic reprogramming via deletion of CISH in human iPSC-derived NK cells promotes in vivo persistence and enhances anti-tumor activity” generated iPSCs-derived NK cells bearing a knockout of the *CISH* gene, which encodes the CIS protein that interferes with IL15 signaling in NK cells. The authors demonstrated enhanced JAK-STAT signaling in IL15-induced CISH−/− iPSC-NK cells. As a consequence, CISH−/− iPSC-NK cells demonstrated better survival in vivo and increased cytotoxic activity against the MOLM-13 multiple myeloid leukemia cell line [73].

Obtaining NK cells from iPSCs seems to be a promising direction for cellular immunotherapy, offering certain advantages over immunotherapy with T lymphocytes. Nevertheless, protocols for the derivation of T lymphocytes from iPSCs have also been developed. The principle of the directed differentiation of iPSCs into T lymphocytes is the same as for NK cells: this approach mimics the processes that take place in vivo. iPSCs can be obtained from various types of somatic cells by dedifferentiation. However, evidence exists that iPSCs retain “epigenetic memory” and are more predisposed to differentiation towards their parental cell type [74,75]. Therefore, to obtain tumor type-specific T-lymphocyte clones, it is desirable to use T-lymphocyte clones with the same specificity as a source of iPSCs. Using iPSCs derived from other cell types may result in T cells of unknown specificity and TCR rearrangements, complicating their large-scale production. The process of differentiation of iPSCs into T cells includes the following steps [75]:Differentiation of iPSCs into mesodermal cells;Differentiation of mesodermal cells to hemogenic endothelium;Generation of hematopoietic precursors;Generation of T-lymphocyte progenitors (CD8α^−^β^−^/CD4^−^, and then CD8α^−^β^−^/CD4^+^);Generation of double-positive CD8α^+^β^+^/CD4^+^ DP-T cells;Derivation of mature single-positive CD8α^+^β^+^ or CD4^+^ SP-T cells.

The total duration of obtaining mature T-lymphocytes from iPSCs ranges from 7 to 10 weeks, hindering their large-scale production. Multiple growth factors and cytokines are required during the different stages of the differentiation sequence. Hematopoietic differentiation is induced by BMP4, VEGF, SCF, and IL3, which promote the development of hemangioblasts in vivo. The generation of hematopoietic progenitors requires Flt3L, IL7, and the NOTCH factors that play the key roles in the generation of T-lymphocyte progenitor cells [74].

## 5. Discussion

iPSCs are a promising source for NK cell-based biomedical cell products (BMCPs). iPSCs have significant advantages over other sources of NK cells, as was mentioned previously: they overcome the problem of selecting a suitable donor for a patient, provide uniform NK cell populations, are suitable for large-scale BMCP production, can be obtained from various somatic cells, can be genetically modified if necessary, and can be used to obtain characterized clonal NK cell lines. Clinical trials that aim to increase the cytotoxic activity of iNK against tumor cells are currently underway. Methods of obtaining genetically modified iNKs with enhanced and specific antitumor toxicity that are safe for the patient are being developed. One of the most intriguing lines of research in this area is the generation of CAR-iNK cells with directed cytotoxicity for a specific tumor antigen and the increased expression of key activating receptors, such as CD16. New protocols for the targeted differentiation of iPSCs into NK cells are being developed, in better accordance with GMP in terms of xeno-free requirements, and with higher yields of iNK cells [76,77].

The protocols considered in detail in the previous section (along with the few variations given in Table 4) show pronounced uniformity in terms of consumables, media, time consumption, verification methods, and most notably, the assumed underlying biology. The main features are as follows:Comprise two key events: (1) differentiation of iPSCs into LPs and (2) further differentiation of the progenitors into NK cells;Total run time ~5–7 weeks: 1–2 weeks to obtain LPs from iPSCs, plus 4–5 weeks to differentiate the progenitors into NK cells;A run yields ~10^6^ cells of the desired CD45+CD56+ phenotype; the exact yields largely depend on the cell culture system (devices) used for the procedure. After expansion (scaling up), the yields may reach a clinically relevant scale of 10^9^, considering the currently established therapeutic dose of 10^7^ cells per kg body weight;The transition to LPs proceeds under the influence of the recombinant human proteins SCF, BMP4, and VEGF; human serum albumin, recombinant or plasma-derived; and a ROCK inhibitor (thiazovivin or Y-27632). The xeno-free requirements are met by particular commercial cell culture products to be used for the purpose, notably APEL™ medium;The transition to NK cells involves recombinant human SCF, IL7, IL15, IL3, and FLT3L, and human serum albumin;The expansion procedure (scaling up) uses culture media supplemented with heat-inactivated human AB serum, IL15, IL21, and IL2; the expansion involves coculture with the K562 tumor cell line (IL15+IL21+, capable of NK cell activation).

Noteworthily, in biomedical NK cell products, the CD56^bright^ phenotype should be prioritized as functionally incremental (compared to the more mature but non-durable CD56^dim^). Almost none of the current protocols specify the characteristic proportion of CD56+ subsets in the yields; this parameter clearly requires optimization, possibly by adjusting the conditions (primarily the time length) used for the differentiation and expansion steps.

General restrictions apply to the use of non-human bioadditives, both natural and artificially obtained (for example, recombinant proteins must correspond to human templates). Accordingly, feeder or stromal cell cultures may not be used at any stage, unless they have been isolated from a human and irradiated with a verified high dose to completely prevent propagation. Also, instead of Matrigel^®^, it is necessary to use humanized equivalents of protein hydrogel coating; as additives, it is permissible to use products of chemical synthesis, inactivated plasmapheresis products, or recombinant proteins (interleukins, etc.) based on human templates.

## 6. Conclusions

Human induced pluripotent stem cells provide a potent source of NK cells for CAR NK cell therapy of various cancers. Apart from their universal availability from any donor (autologous or allogeneic), iPSCs can be flexibly manipulated toward enrichment with functionally valuable phenotypes, e.g., the CD56^bright^ population of NK cells. The feeder-free contemporary protocols are promising in terms of clinical implementation, albeit needing to achieve higher compliance with the xeno-free requirements prior to being established as a standard.

## Figures and Tables

**Figure 1 ijms-26-01107-f001:**
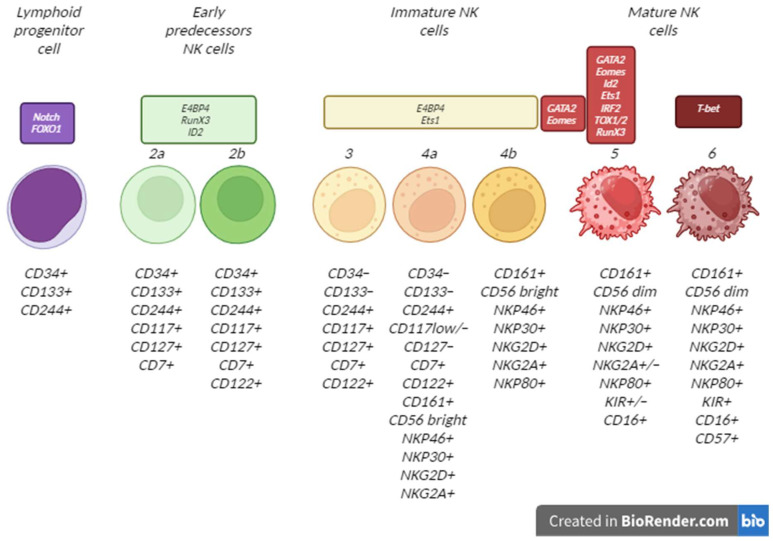
Consecutive stages of NK cell differentiation from LPs, with characteristic surface markers and transcription factors involved in differentiation and maturation.

**Figure 2 ijms-26-01107-f002:**
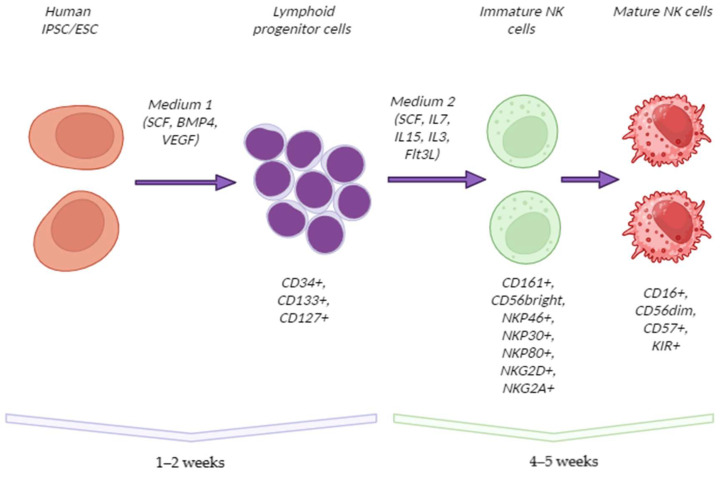
A scheme of iPSC differentiation into NK cells (created with BioRender.com).

**Table 1 ijms-26-01107-t001:** Registered clinical trials of iPSC-derived CAR NK cell products.

#	ID	Cell Product	Genes Targeted *	Phase **
1	NCT04245722	FT596	CAR, *hnCD16*, *IL15RF*	1
2	NCT04555811	FT596	CAR, *hnCD16*, *IL15RF*	1
3	NCT05182073	FT576	CAR, *hnCD16*, *IL15RF*, *CD38*^KO^	1
4	NCT05395052	FT536	CAR, *hnCD16, IL15RF*, *CD38*^KO^	1
5	NCT05336409	CNTY-101	CAR, *dIL-15*, *HLA-I*^KO^, *HLA-II*^KO^, *HLA-E*^KI^, *EGFR*	1

* modified, introduced, or inactivated; KO, knockout; KI, knock-in. ** as of 2023.

**Table 2 ijms-26-01107-t002:** Transcription factors involved in NK cell differentiation and maturation.

Differentiation	Maturation
E4BP4	JAK-1/3–STAT5
TCF1	T-bet
ETS1	EOMES
ID2	TOX
NOTCH	ID2
	TOX/TOX2
	PRDM1
	ZEB2
	GATA3
	SMAD4

**Table 3 ijms-26-01107-t003:** NK surface marker signatures at consecutive stages of differentiation from LPs.

Stage/ Surface Receptor Signatures	LP	EP-NK	LP-NK	IM-NK (1)	IM-NK (2)	IM-NK (3)	M-NK (1)	M-NK (2)
CD34	+	+	+	-	-	-	-	-
CD133	+	+	+	-	-	-	-	-
CD244	+	+	+	+	+	+	+	+
CD117	-	+	+	+	low/-	low/-	low/-	-
CD127	-	+	+	-	-	-	-	-
CD7	-	+	+	+	+	+	+	+
CD16	-	-	-	-	-	-	+	+
CD57	-	-	-	-	-	-	-	+
CD122	-	-	+	+	+	+	+	+
NKP46	-	-	-	-/+	+	+	+	+
NKP30	-	-	-	-/+	+	+	+	+
CD161	-	-	-	-/+	+	+	+	+
CD56	-	-	-	-	bright	bright	dim	dim
NKP80	-	-	-	-	-	+	+	+
KIR	-	-	-	-	-	-	+/-	+
NKG2D	-	-	-	-	+	+	+	+
NKG2A	-	-	-	-	+	+	+/-	+/-

LP, Lymphoid progenitors; EP-NK, Early NK cell precursors; LP-NK, Late NK cell precursors; IM-NK (1), Immature NK cells (stage 3); IM-NK (2), Immature NK cells (stage 4a); IM-NK (3), Immature NK cells (stage 4b); M-NK (1), Mature NK cells (stage 5); M-NK (2), Mature NK cells (stage 6). -, lack of receptor expression; +, presence of receptor expression; low, low expression of the receptor; +/-, -/+, the expression of the receptor is either found or absent, the first sign indicates the frequency of occurrence, if -, then it is rare, if +, then it is regularly.

## Data Availability

The datasets used and/or analyzed during the current study are available from the corresponding author upon reasonable request.

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
