# Peer review of "Differentiating Induced Pluripotent Stem Cells into Natural Killer Cells for Adoptive Cell Immunotherapies—Comparative Characterization of Current Protocols"

_ijms, 2025, doi:10.3390/ijms26031107_

Round 1
Reviewer 1 Report
Comments and Suggestions for Authors
This review article brings different methodological approaches to Natural 2 Killer Cells for adoptive cell immunotherapies widely tested under in-vivo conditions and explains the methodology and output in detail. It is an interesting review article on this domain. Before being recommended for possible consideration for publication, it would be nice to emphasize the following suggestion to enrich the quality of the manuscript. Appreciate the authors for providing clear multiple versions of tables to give quantitative information about on going research output.
1. The main objective of this review needs to be classically narrated in the introduction section. It seems like the authors write to express their views on this topic. Do they want to express the technology gap between NK cells in real-time applications?
2. In conclusion or before the discussion and conclusion, the author needs to emphasize, what would be a future perspective on this NK system on medical realization, since the author cited accountable references (read much understanding). From this perspective, it would be nice to give some ideology combined with bottleneck challenges and potential opportunities in this research directions
3. Finally, would like to see the overall outcome from this review will be more supportive for future readers to understand the on-going approaches.
Author Response
Comments and Suggestions for Authors
This review article brings different methodological approaches to Natural 2 Killer Cells for adoptive cell immunotherapies widely tested under in-vivo conditions and explains the methodology and output in detail. It is an interesting review article on this domain. Before being recommended for possible consideration for publication, it would be nice to emphasize the following suggestion to enrich the quality of the manuscript. Appreciate the authors for providing clear multiple versions of tables to give quantitative information about on going research output.
- The main objective of this review needs to be classically narrated in the introduction section. It seems like the authors write to express their views on this topic. Do they want to express the technology gap between NK cells in real-time applications?
Response 1: Thank you very much, we have edited the text and formulated the aim in the last paragraph of the introduction (lines 35-38). The goal of our review is to compare the existing protocols for obtaining the NK cells from iPSCs, since natural killers are considered as a promising approach for cancer immunotherapy. We outline key principles of directed differentiation of iPSCs into NKs and assess the compliance of these protocols with Good Manufacturing Practices (GMP) standards in xeno-free conditions. The fact is that there is no unified protocol for generating NKs from iPSCs to date. Some protocols for obtaining NK cells from iPSCs discussed in this review do not comply with GMP conditions, while the others, on the contrary, are promising for using in the clinical practice. Also, the presence of clinical trials registered on the ClinicalTrials.gov website evidences for intense optimization of methods for obtaining NK cells from iPSCs that meet GMP standards in terms of xeno-free requirements.
- In conclusion or before the discussion and conclusion, the author needs to emphasize, what would be a future perspective on this NK system on medical realization, since the author cited accountable references (read much understanding). From this perspective, it would be nice to give some ideology combined with bottleneck challenges and potential opportunities in this research directions.
Response 2: Thanks to your recommendation we have formulated several additional conclusions about the prospects for using the obtained iNK cells and future perspective of this area (lines 543-556)
- Finally, would like to see the overall outcome from this review will be more supportive for future readers to understand the on-going approaches.
Response 3: Thank you for your recommendation. We have supplemented the discussion section with the main aspects of using NK cells obtained from iPSCs for cellular immunotherapy. Some of the main approaches to obtaining NK cells from iPSCs are presented in Table #4, the general principles of obtaining are described in Section 4. We also provided some clinical trials of iNK cells in the treatment of various types of tumor diseases (lines 488-513).

Reviewer 2 Report
Comments and Suggestions for Authors
The review paper entitled “Differentiating Induced Pluripotent Stem Cells into Natural Killer Cells for Adoptive Cell Immunotherapies — comparative characterization of current protocols” by Budagova et al. is very well written, and the authors compared different protocols for differentiation of iNK from iPSCs. The authors have discussed the 3 existing protocols and clearly explained the advantages and disadvantages of each methodology. However, they did not address which of these protocols better induce the survivability of NK cells in vivo. Additionally, while several protocols have been designed and published, none of the current methods appear to successfully generate highly mature iNK cells with enhanced in vivo survivability. Furthermore, the authors did not explore how manufacturing NK cells from iPSCs compares complexity to T cell differentiation. Including a brief comparison between T and NK cells would be highly valuable for readers in the field. I would suggest the authors include the abovementioned suggestions in the discussion section.
Author Response
Comments and Suggestions for Authors
The review paper entitled “Differentiating Induced Pluripotent Stem Cells into Natural Killer Cells for Adoptive Cell Immunotherapies — comparative characterization of current protocols” by Budagova et al. is very well written, and the authors compared different protocols for differentiation of iNK from iPSCs. The authors have discussed the 3 existing protocols and clearly explained the advantages and disadvantages of each methodology.
- However, they did not address which of these protocols better induce the survivability of NK cells in vivo. Additionally, while several protocols have been designed and published, none of the current methods appear to successfully generate highly mature iNK cells with enhanced in vivo survivability.
Response 1: Thank you very much for your interesting suggestions. We have indicated at the end of each Method that no in vivo work was performed within the framework of the studies conducted. In addition, we have provided data from Dan S Kaufman's papers on assessing iNK survival in vivo (lines 448-513).
- Furthermore, the authors did not explore how manufacturing NK cells from iPSCs compares complexity to T cell differentiation. Including a brief comparison between T and NK cells would be highly valuable for readers in the field. I would suggest the authors include the abovementioned suggestions in the discussion section.
Response 2: We have included in the review a brief description of the differentiation of iPSCs into T-lymphocytes and outlined the main difficulties in obtaining them (lines 516-540).
